# Effectiveness of artificial urinary sphincter to treat stress incontinence after prostatectomy: A meta-analysis and systematic review

**Yue Li**[1], **Xiao Li**[1]*, **Qin Yang**[2]

1 Nursing College of Yunnan University of Traditional Chinese Medicine, Kunming City, Yunnan province, China, 2 College of Nursing Dali University, Dali City, Yunnan province, China

* 1877661038@qq.com

**Data Availability Statement:** All relevant data are within the paper and its Supporting information files.

## Abstract

### Background

Artificial Urinary Sphincter (AUS) has always been considered the gold standard for surgical treatment of male non-neurogenic Stress Urinary Incontinence (SUI). The purpose of this meta-analysis was to evaluate AUS's effectiveness in treating male SUI, as described in the literature.

### Methods

Two independent reviewers used PubMed, EMBASE, Web of Science, CNKI, WanFang Data, and VIP databases, to find the efficacy of artificial urethral sphincter in treating SUI after male prostate surgery. We excluded studies on female urinary incontinence. The main purpose of this study was to evaluate the clinical efficacy based on the degree of dry rate after AUS AMS 800™: postoperative complete dry was defined as no pad use per day. Post-operative social dry was defined as 0–1 pad per day. The secondary goal was to analyze the use of AUS AMS 800™ to improve SUI and to calculate the degree of influence by analyzing the number of pads and postoperative quality of life. And methodologic quality of the overall body of evidence was evaluated using the GRADE (Grading of Recommendations Assessment, Development, and Evaluation) guidelines.

### Results

The data in this paper are mostly based on prospective or retrospective cohort studies without control groups. Fortunately, most studies have the same criteria to assess effectiveness. The pooled data of 1271 patients from 19 studies (6 prospective cohort studies, 12 retrospective cohort studies, and 1 randomized controlled trial) showed that: the number of pads used (pads/ day) after AUS was significantly reduced by about 4 (P < 0.001) and the quality of life was improved (P < 0.001).In addition, data analysis showed a high degree of heterogeneity between studies. According to the severity of baseline SUI, subgroup analysis was performed on the postoperative dry rate and social dry rate. Although heterogeneity was reduced, $I^2$ is still above 50%, considering that

**Funding:** The author(s) received no specific funding for this work.

**Competing interests:** The authors have declared that no competing interests exist.

heterogeneity may not be related to the severity of SUI. The random effect model was used for data analysis: the dry rate was about 52% (P < 0.001), and the social dry rate was about 82% (P < 0.001). The evidence level of GRADE of dry rate is very low, the evidence level of social dry rate and Pads use (pads/day) is Moderate, and the evidence level of Quality of life is low.

## Conclusion

Although the evidence in this paper is based on descriptive studies and limited follow-up, the results show that AUS is effective in treating urinary incontinence and can improve patients' quality of life.

## Introduction

The treatment of post-prostatectomy incontinence (PPI) is still a challenge for urologists and their patients after prostate surgery. Despite the continuous improvement of surgical techniques, SUI is one of the sequelae that have the greatest negative impact on the quality of life of patients after prostate surgery (mainly including radical prostatectomy, open prostatectomy, transurethral resection of the prostate, transurethral prostatectomy, transurethral laser/vaporization/enucleation of the prostate) [1, 2]. Sphincter dysfunction caused by postoperative sphincter injury or weakness is considered to be the most important cause of persistent SUI after prostatectomy. Postoperative SUI will reduce the prognosis of patients, resulting in disease shame, being far away from social contact, and social disengagement [3], which greatly affects patients' quality of life after surgery.

The preferred treatment for postoperative SUI mainly includes pelvic floor muscle training and drug therapy. However, for the negative impact of continuous SUI on the quality of life of postoperative patients, surgery is still the only choice for active treatment. Since 1972, AUS has become the treatment standard for severe SUI caused by internal sphincter dysfunction, which has completely changed the treatment of male SUI [4]. In 1987, the release of the AMS 800™ device (Boston Scientific, Boston, USA) marked the maturity of the device [5]. The device comprises an inflatable cuff placed around the urethra, a pressure-regulating balloon that keeps the cuff inflation, and a pump placed in the scrotum. The male squeezes the pump to achieve urination. The purpose is to close the urethra and dry the patient. Once implanted, the device is deactivated in an open position for 4–6 weeks so that the postoperative swelling subsides, and then the device is clinically activated for use. The success rate of AUS in treating SUI after prostate surgery is as high as 79%, which is considered the gold standard for treating male SUI [1]. It is estimated that more than 150,000 patients worldwide have been implanted with AUS. This significant number of clinical cases may have a very long follow-up time, but it is almost not reflected in the literature [6]. The definition of inclusion and exclusion criteria in many kinds of literature is unclear, and even patients with SUI due to different causes are included. Therefore, we aim to conduct a meta-analysis to evaluate the effectiveness and safety of AUS in the management of post-prostate surgery SUI. This analysis aims to evaluate the current evidence on the effectiveness and safety of AUS in treating SUI after male prostate surgery.

## Methods

### Search strategy

According to the PRISMA guidelines, [7] a meta-analysis was conducted by systematically searching Chinese databases including CNKI, WanFang Data, and VIP, as well as foreign

language databases such as Cochrane Library, Web of Science, Embase, PubMed, and Clinical-Trials.gov (the search time was from the establishment of the database to March 20, 2023). Subject words were combined with free word search, and search strategies were developed according to PICOS standards (population, intervention, control, outcome, and study design) (Table 1). The search uses the following terms: "urinary sphincter, artificial", "artificial urinary sphincter", "Urinary Sphincters, Artificial"; "stress urinary incontinence";" male". Potential studies were manually searched using the snowball method. Two researchers independently and thoroughly identified, selected, and extracted data from the studies. The study was selected for the first time by reading the title and abstract, followed by a full-text review of studies that met the inclusion criteria. Differences were resolved by consensus or in collaboration with a third research team member.

The included studies encompassed randomized controlled trials, prospective cohort studies, and retrospective cohort studies. All included studies evaluated the efficacy and safety of AUS for patients with SUI after prostate surgery. The literature includes only English and Chinese literature.

## Data extraction and quality assessment

The following information was extracted from the studies that met the inclusion criteria: first author's name, publication year, study design, demographic data of the subjects, intervention type, follow-up time, and outcome before and after intervention.

The quality of included studies was assessed using a modified 18-item Delphi checklist [8]. The tool aims to assess the quality of non-comparative studies. The quality assessment was conducted by two reviewers, and any discrepancies were resolved through consultation with a third reviewer.

## Statistical analysis

The primary efficacy indicators of the meta-analysis were dry rate (defined as patients using 0 pads per day) and social dry rate (defined as patients using 0–1 pads per day). Secondary efficacy indicators were daily use of pads and quality of life before and after surgery. Due to the lack of a control group, all outcomes were tested by comparing the follow-up data with the baseline data (i.e., within-group effects). The collected data of dry rate and social dry rate were converted into standard errors, and 95% confidence intervals (CIs: lower and upper limits) were used for statistical evaluation. For daily use of pads and quality of life, the mean value with standard deviation was calculated and compared before and after surgery. However, if the study did not report the mean and standard deviation, the mean and standard deviation were estimated from the sample size, median, range, or interquartile range [9]. Suppose a study reports the median and interquartile range (IQR), we assume that the median of the outcome variable is equal to the mean effect, with an IQR width of approximately 1.35 standard

**Table 1. PICOS criteria to guide the meta-analysis.**

| Population | Male patients with mild, moderate, or severe stress urinary incontinence after prostatectomy |
|---|---|
| Intervention | artificial urinary sphincter AMS 800™ (Boston Scientific, Boston, USA) |
| Comparison | None available |
| Outcomes | Primary: complete dry rate (0 pads/day); Social dry rate(0~1 pad/day) |
| | Secondary: differential pad count (after adjustment with respects to baseline), Quality of life |
| Study design | RCT, Retrospective, and prospective cohort studies |

deviations [10]. Due to the different measurement tools used for the continuous variables in the included literature, this study employed standardized mean difference (SMD) as the pooled effect size. Stata 17.0 software was used for meta-analysis of the final included research data. Firstly, the heterogeneity test of each research result is carried out. If $I^2 < 50\%$, the heterogeneity is acceptable, and the fixed effect model is used; If $I^2 > 50\%$, indicating high heterogeneity, using the random effects model; If $I^2 > 75\%$, which indicates high heterogeneity, a random effects model is used. The funnel plot was used to assess publication bias, with symmetrical distribution among the indicators suggesting the absence of publication bias. In the Egger test, the results of the outcome indicators were quantitatively tested. When $P < 0.05$, there may be a greater possibility of publication bias. The leave-one-out approach was used to analyze the sensitivity of the main outcomes.

## Studies quality

GRADE (Grading of Recommendations Assessment, Development and Assessment). The assessment involves within-study risk of bias, directness of evidence, inconsistency of effect estimates (heterogeneity), precision of effect estimate and risk of publication bias. Confidence of the effect estimates was described as high, moderate, low, and very low (Table 2).

## Results

### Literature search

A total of 19 studies were included, including 6 prospective cohort studies [11–16], 12 retrospective cohort studies [17–28], and a randomized controlled trial (RCT) [29]. The initial search identified 539 studies, and the topics of the initial search literature were imported into Note Express. After checking the duplicates, 162 same kinds of literature were removed. Then, by reading the title and abstract, 345 articles that did not meet the inclusion criteria were excluded, and 32 were left. 6 studies were screened by the snowball method, and 38 studies were comprehensively reviewed (Fig 1). Among the 38 studies, 19 studies were excluded for various reasons: no quantified outcome was reported (n = 3), the subjects were female patients (n = 2), neurogenic urinary incontinence (n = 2), surgical equipment was not AUS AMS 800™ model (n = 4) review (n = 2), case (n = 3), technical report (n = 1), non-Chinese and English literature (n = 2).

These studies included 1271 patients. The primary intervention technique was single / double-cuff AUS. The average age of patients undergoing AUS implantation ranged from 65 to 78 years. Median follow-up was 34.6 months for AUS. In all studies, the number of pads used daily after surgery decreased. The dry rate ranges from 7.3% to 80% and the social dry rate ranges from 59% to 100%. Overall, the quality of life improved after the AUS intervention. The lost to follow-up rate was between 2.6% and 16%. The complication rate was 12.5% to 50%.

**Table 2. GRADE working group grades of evidence.**

| Quality of evidence | Interpretation |
| --- | --- |
| High | Further research is very unlikely to change our confidence in the estimate of effect. |
| Moderate | Further research is likely to have an important impact on confidence in the estimate of effect and may change the estimate. |
| Low | Further research is very likely to have an important impact on confidence in the estimate of effect and is likely to change the estimate |
| Very low | Any estimate of effect is very uncertain. |

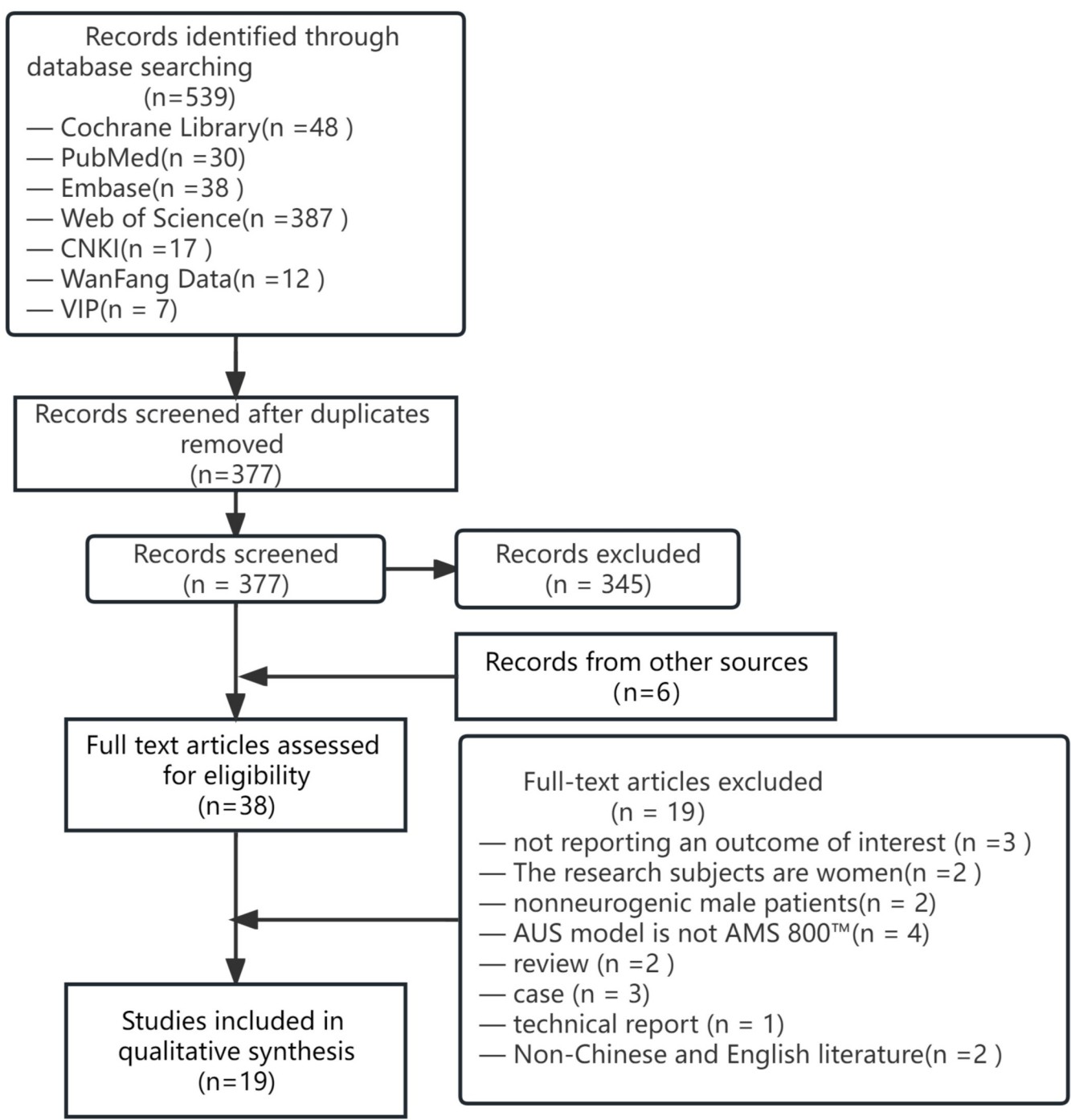

**Fig 1. Study selection flow chart.**

Patients with a history of UI surgery accounted for 20% to 73%, while 4.3% to 48% of patients received radiation therapy. Patients with re-do procedures ranged from 2% to 21% (Table 3).

## Quality assessment of included studies

The quality assessment results are summarized as follows: The quality of included studies was evaluated based on 18 items from the Delphi checklist. If the literature≥14 items of the Delphi checklist, it was considered to meet acceptable quality criteria [30]. In this study, 11 studies evaluating AUS met≥14 "Yes" answers, indicating that their quality was qualified [13–15, 17 19, 21–23, 25, 28]. The criteria lacking in the study include reports of additional interventions, several studies being non-multicenter studies, patients being recruited non-consecutively, failure to report lost to follow-up rates. The Quality assessment of included studies is presented in Additional S1 Table.

## Meta-analysis results

**Meta-analysis of dry rate.**   Eighteen studies [11–19, 21–29] reporting dry rates after AUS placement were included and included in the meta-analysis. Significant heterogeneity was observed among the studies based on the P and $I^2$ values ($I^2$ = 97.0%, P = 0.000). When comparing studies with a mean baseline pad count of≥6 pads/day (severe SUI) to studies with <6 pads/day (mild-moderate SUI), the results showed substantial heterogeneity in both groups ($I^2$ = 95.3% for ≥6 pads/day group, $I^2$ = 79.4% for <6 pads/day group), indicating no significant improvement in heterogeneity. Therefore, a random-effects model was used for the meta-analysis, and the results showed a statistically significant difference in dry rate among male patients after AUS surgery, with a rate of 52% (95% CI = 0.39–0.66) (Fig 2).

**Meta-analysis of social dry rate.**   A meta-analysis included 18 studies [11–20, 22–29] that reported postoperative social dry rate. According to P and $I^2$ values ($I^2$ = 93.7%, P = 0.000), studies have heterogeneity. Similarly, subgroup analysis was performed according to the severity of urinary incontinence. The results showed that urinary pad count ≥ 6pads / day group, $I^2$ = 95.3%; urinary pad count < 6pads / day group, $I^2$ = 81.4%, Heterogeneity was not significantly improved. Therefore, a random effect model was used for meta-analysis. The results showed that the social dry rate of male patients after AUS was 81% (95% CI = 0.73–0.89), and the difference was statistically significant (Fig 3).

**Meta-analysis of postoperative daily pad usage.**   9 of the 19 studies [13, 20–22, 24, 25, 27–29] provided data on patients ' daily pad use before and after surgery and were included in the meta-analysis. There was evidence of heterogeneity among the 9 studies ($I^2$ = 92.8, P = 0.000). Therefore, the random effect analysis model is used. Due to the different measurement tools used for continuous variables, This study used standardized mean difference (SMD). The mean difference of comprehensive SMD was (2.68,95% CI = 2.07–3.29). It showed that the daily use of pads was significantly reduced after surgery (P < 0.001) (Fig 4).

**Meta-analysis of postoperative quality of life.**   7 studies [12, 17, 22, 23, 25, 27, 29] provided data on patients' quality of life before and after AUS.3 were ICIQ-SF scores, and the remaining 4 were VAS scores. Subgroup analysis was performed according to different scoring criteria. There was heterogeneity in the ICIQ-SF score group ($I^2$ = 97.3%, P = 0.000); VAS score group, ($I^2$ = 92.3%, P = 0.000). Therefore, a random effect model was used for meta-analysis due to the different measurement tools used for continuous variables. This study used standardized mean difference (SMD). The combined effect size results show that: In the ICIQ-SF score group, the mean difference of comprehensive SMD was (1.77,95% CI = 1.57–1.96); in the VAS score group, the mean difference of comprehensive SMD was (3.45,95%

**Table 3. Summary of basic characteristics and outcomes of selected studies for meta-analysis.**

| 1at Author (year) | Number of patients | Age (years) | Mean follow-up (months) | Definition of dry rate (%) | Dry rate (%) | Definition of Social dry (%) | Social dry rate (%) | Pads use (pads/day) Pre vs. Post (PPD) | Quality of life (Pre vs. Post) | Study Design | intervention technique | lost to follow-up | Complication rate (%) | History of UI surgery (%) | Radiation (%) | re-do procedures (%) |
|---|---|---|---|---|---|---|---|---|---|---|---|---|---|---|---|---|
| Sacco (2021) | 35 | 71 (67–74)² | NA | 51.2 (32–62.2)² | 0 pads/day | 62.9 | 0–1 pad/day | 91.4 | 4 (4–5)²vs.0 (0) | ICIQ score 17 (15–18)²vs.4 (0–6)² | R | ① | 5.7 | 20 | 25.7 | 25.7 | NA |
| Kuznetsov (2000) | 36 | NA | NA | 0 pads/day | 33 | ≤1 pad/day | 75 | NA | NA | NA | R | ② | 12 | 0 | 20 | NA | 2.7 |
| Imamoglu (2005) | 11 | 64 (52–76)¹ | 12 | 0 pads/day | 72.7 | ≤1 pad/day | 90.9 | 2.27vs.0.36 | Quality of life scale33.3vs.9.2 | P | ② | NA | NA | 0 | 0 | NA |
| Ahyai (2016) | 157 | 70 (65.0–73.5)² | 24 (9–33)² | 0 pads/day | 79 | 0–1 pad/day | 93 | 7 (5–8)²vs.1 | NA | R | ③ | 7 | 20.6 | 34.1 | 31.4 | 7.8 |
| Sotelo (2008) | 83 | 67.6 ± (8.7) | 18.8 ± (14.6) | NA | NA | ≤1 pad/day | 83 | 6.7±4.0vs.1.1 ±1.6 | NA | R | ③④ | NA | 33.6 | NA | 29 | 17 |
| Serra (2017) | 82 | 68(54–78)³ | 46(12–135)³ | 0 pads/day | 76 | 0–1 pad/day | 92 | NA | ICIQ score 19 (8–21)²vs.4(0–17)² | P | ③⑤ | NA | 17 | 0 | 5.9 | 4.9 |
| Grabbert (2019) | 220 | 70 | 16.7 ±14.81 | NA | 57.3 | NA | NA | 6.87± 3.98vs.1.04 ±0.03 | NA | R | NA | 15 | NA | 36.7 | 43.1 | 9.1 |
| Fan Zhang (2022) | 12 | 68.5 ±6.5 | 54(8–120)³ | 0 pads/day | 66.7 | ≤1 pad/day | 83.3 | 3.9±1.4vs.1.1 ±1.1 | VAS score 8.3 ±1.0vs.2.7±1.2 | R | ③⑥ | 0 | 16.7 | 25 | 16 | NA |
| Trigo (2008) | 40 | 68.3 ±6.3 | 53.4 ±21.4 | 0 pads/day | 50 | ≤1 pad/day | 90 | 4.0 ±0.9vs.0.62 ±1.07 | NA | P | NA | 0 | 12.5 | NA | 0 | 20 |
| Fan Zhang (2018) | 5 | 47.2 ±24.5 | NA | ≤ 1 pad/day | 80 | ≤ 2 pads / day | 100 | NA | VAS score 8.3 ±0.6vs.2.3±0.7 | R | ③⑦ | 0 | NA | NA | 0 | 40 |
| Lin-Feng Meng (2019) | 5 | 75.4 (71–78)³ | 28.4(3–60)³ | NA | 20 | ≤2 pads / d | 80 | 2.4 ±0.89vs.1.4 ±1.14 | NA | R | ③⑧ | 0 | 40 | 0 | NA | 20 |
| Fan Zhang (2022) | 46 | 45.6 ±16.0 | 85(6–228)³ | 0 pads/day | 34.8 | ≤1 pad/day | 76.1 | 3.5 ±1.05vs.1.2 ±0.65 | VAS score 7.1 ±1.2vs.2.6±1.9 | R | ③⑦ | 0 | 32.6 | 78.2 | 4.35 | 6.5 |
| Maurer (2019) | 219 | 70.0 (65.0–74.0)² | 24(6–31)² | 0 pads/day | 75.8 | <2 pads/day | 87 | 7 (5–8)²vs. NA | NA | P | ③ | NA | 33.3 | 29.7 | 31.5 | NA |
| Maure (2020) | 150 | 70.0 (66.0–74)² | 24 (7.25–36)² | 0 pads/day | 77.3 | <2 pads/day | 86 | 7 (5.75–8.25)²vs.NA | NA | P | ③⑨ | 8 | 48.6 | NA | 48.7 | NA |
| O'Connor (2008) | 41 | 67 | 66 | 0 pads/day | 7.3 | 0–1 pads/day | 59 | 7.7vs.1.3 | NA | R | ③ | 16 | 19 | NA | NA | 17 |
| Constable (2022) | 190 | 68 ±6 | NA | NA | 15.8 | less than once a week and a small amount | 34.8 | 3.7 ±2.2vs.1.2 ±1.2 | ICIQ score 16.4 ±3.2vs.7.1 ± 5.0 | RCT | NA | 2.6 | 50 | 0 | 20 | 2 |
| Fan Zhang (2016) | 17 | 40.29 ±14.78 | 40.8(13–144)³ | NA | 47.1 | <2pads/day | 82.3 | 3.62 ±0.35vs.1.29 ±0.31 | VAS score 6.75 ±0.36vs.1.86 ±0.6 | R | ③⑩ | 0 | 26.7 | 73.3 | NA | 13.3 |
| Mottet (1998) | 96 | NA | NA | 0 pads/day | 61 | 0–1 pads/day | 92 | NA | NA | P | NA | NA | NA | NA | NA | 21 |

(*Continued*)

**Table 3.** (Continued)

| 1st Author (year) | Number of patients | Age (years) | Mean follow-up (months) | Definition of dry rate (%) | Dry rate (%) | Definition of Social dry (%) | Social dry rate (%) | Pads use (pads/day) Pre vs. Post (PPD) | Quality of life (Pre vs. Post) | Study Design | intervention technique | lost to follow-up | Complication rate (%) | History of UI surgery (%) | Radiation (%) | re-do procedures (%) |
|---|---|---|---|---|---|---|---|---|---|---|---|---|---|---|---|---|
| O'Connor (2007) | 33 | 77.6 (75–83)[1] | 60(12–72)[1] | 0 pads/day | 24 | one half to one pad daily | 83 | 6.7 (3–10)[1] vs.0.8 (0–2)[1] | NA | R | ⑪ | 12.1 | 45 | NA | 38 | 14 |

[1] . mean (range).

[2] . median (IQR).

[3] .median (range).

[4] . NA, not available.

① abdominal -perineal surgical approach, ② implanted around the proximal bulbar urethra, ③ single/double-cuff, ③④single-incision, transverse scrotal approach, ③⑤ perineal, penoscrotal, and preserving the bulbospongiosus muscle approach, ③⑥Transperineal double /single incision; Scrotum single incision, ③⑦ Transperineal single incision/Scrotum single incision, ③⑧ Place AUS through a single incision, ③⑨ distal bulbar double cuff, ⑩ Perineal and inguinal double cuff /Scrotal single cuff /Scrotal single cuff AUS.

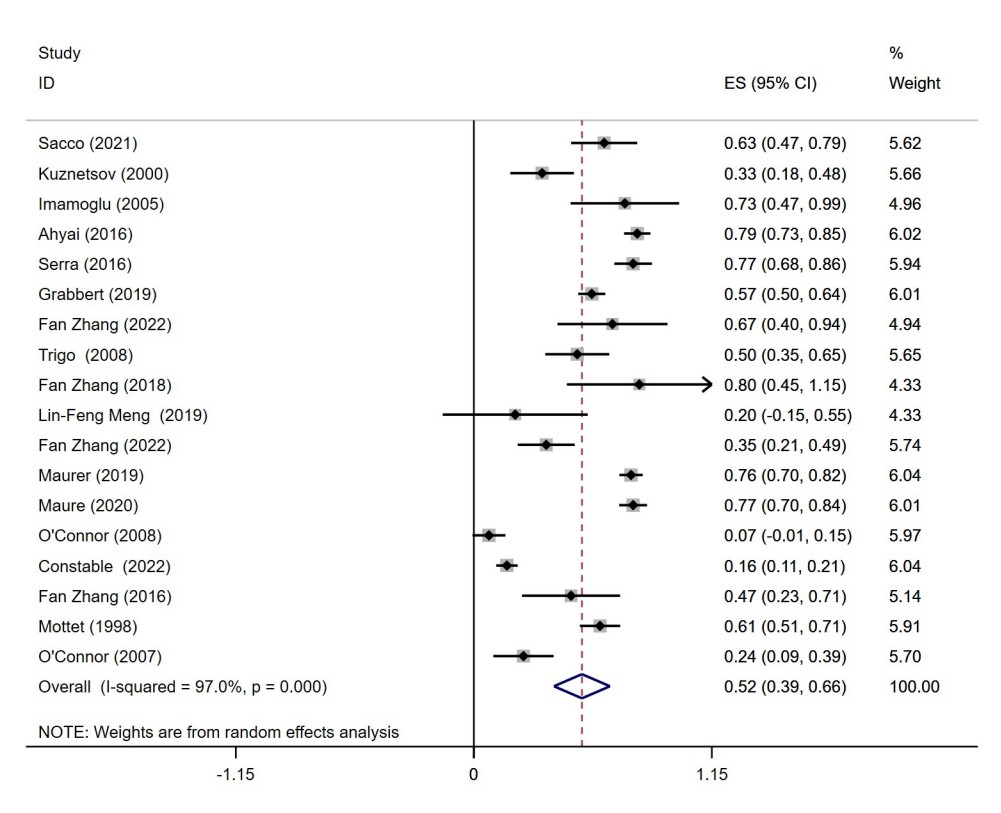

**Fig 2. Meta-analysis for definition of dry rate n (%) for patients treated with AUS.**

CI = 2.92–3.99). The data of both groups showed that AUS could significantly improve patient's quality of life (P < 0.001) (Fig 5).

## Publication bias

According to the Cochrane Handbook criteria, due to the limited number of literature sources (less than 10) available for daily pad usage and quality of life data, it was not possible to conduct a biased assessment using a funnel plot test. Therefore, only the dry rate and social dry rate were assessed for publication bias by creating funnel plots and conducting Egger's test. In this study, the distribution of each index in the dry rate funnel plot was asymmetric, and the Egger test P = 0.04 < 0.05, suggesting that there may be bias (Fig 6). The severity of SUI, the different proportion of patients after previously failed anti-incontinence devices, and the different proportion of patients receiving radiation likely explain the publication bias situation observed. The symmetry of the funnel plot in social dry rate is good, Egger test P = 0.376 > 0.05, suggesting that the possibility of publication bias is low (Fig 7).

## Sensitivity analysis

Sensitivity analysis was performed on the dry rate (Fig 8) and the social dry rate (Fig 9) using the leave-one-out approach, and each study was deleted in turn. The direction and magnitude of the combined estimates did not change significantly with the deletion of any particular

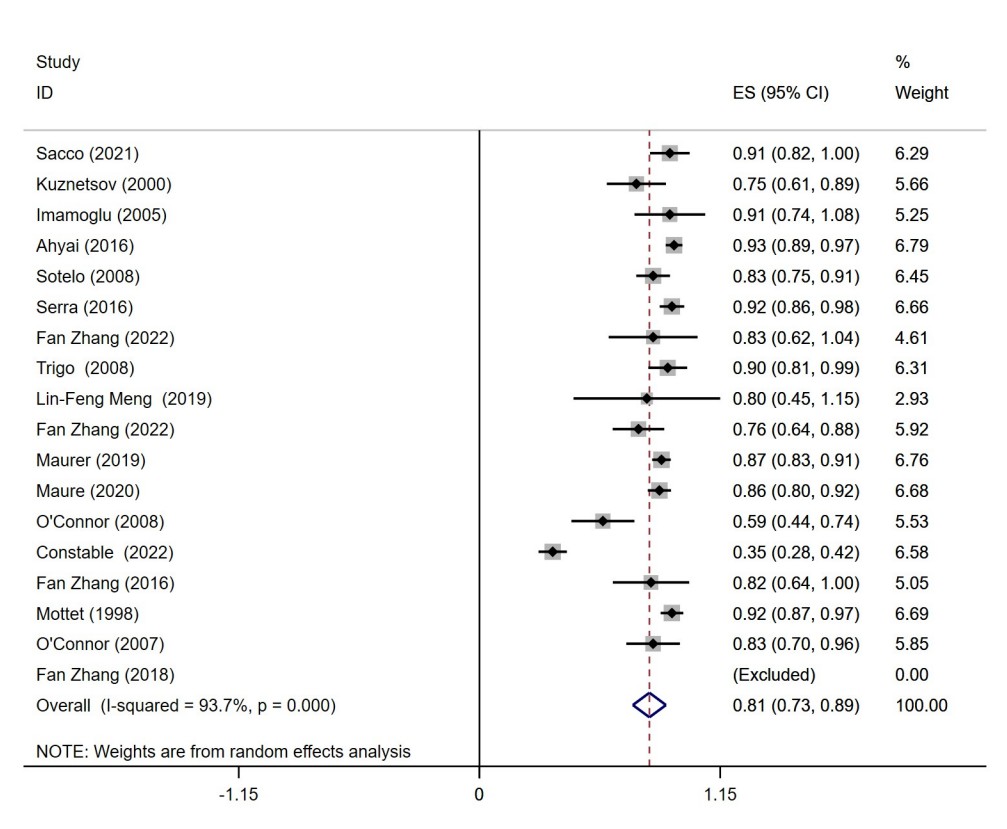

**Fig 3. Meta-analysis for Definition of social dry rate n (%) for patients treated with AUS.**

study. It shows that the meta-analysis has good reliability and the data is not overly affected by any study.

## Rating the quality of evidence

The evidence level of GRADE of Dry rate is very low, the evidence level of Social dry rate and Pads use (pads/day) is Moderate, and the evidence level of Quality of life is low. All studies reported outcome indicators directly. The reasons for the downgrade are as follows (Table 4).

## Discussion

We observed a significant reduction of approximately 4 pads per day in the usage of urinary pads after AUS placement compared to preoperative levels. Moreover, there was a noticeable improvement in the patient's quality of life following the surgery. These findings indicate that AUS treatment methods for urinary incontinence following prostate surgery are effective in enhancing patients' quality of life.

### Discussion of meta-analysis results

**Dry rate and social dry rate.** AUS is the gold standard for the treatment of SUI after prostate surgery. Data analysis of AUS in treating urinary incontinence after radical prostatectomy

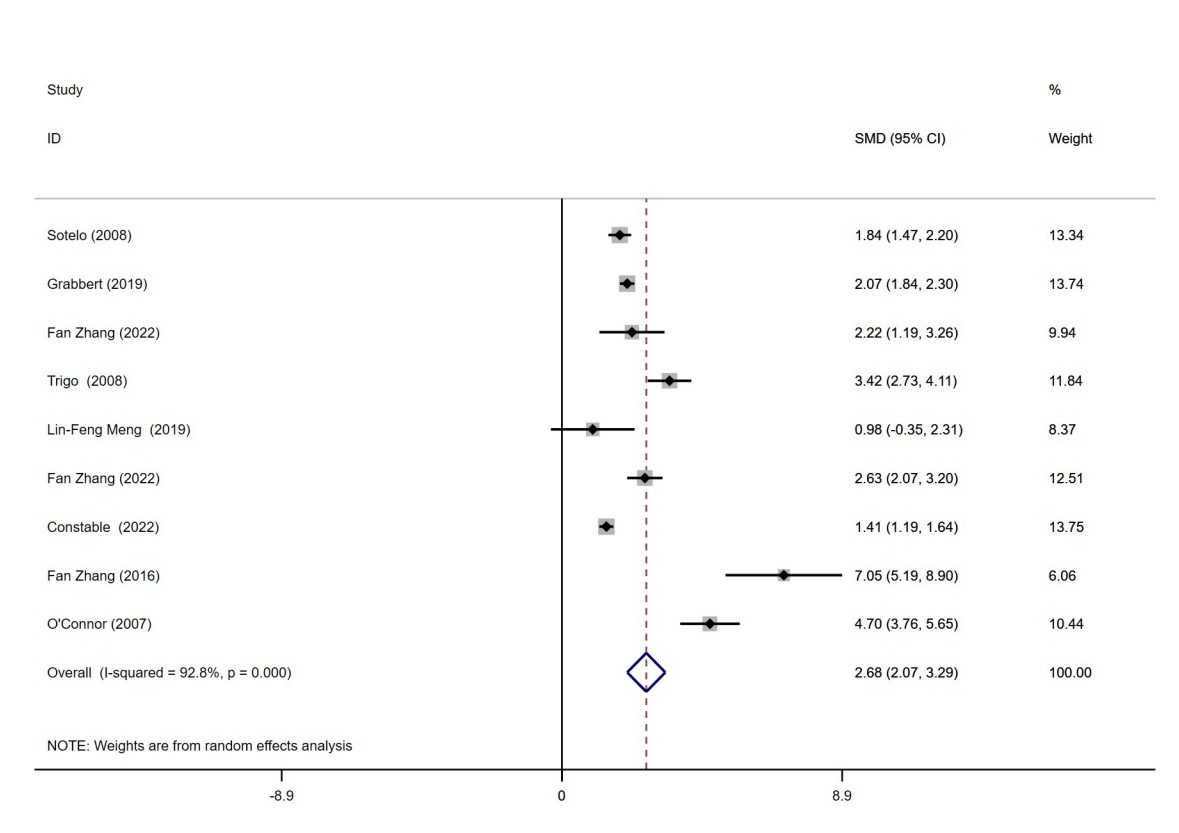

**Fig 4. Meta-analysis for daily pad amount for patients treated with AUS.**

[31] evaluated the therapeutic effects of 8 studies, The follow-up time was about 19 to 34.1 months. Studies have shown that regardless of the degree of urinary incontinence, the improvement rate of postoperative urinary incontinence is about 80%, Consistent with the analysis of this study. This meta-analysis analyzed the data of AUS in the postoperative dry rate and social dry rate of the prostate [11–29]. The study found that the dry rate (0 pads/day) was about 52% after surgery. The social dry rate (0–1 pad/day) was 82%. It shows that the improvement of stress urinary incontinence in about half of the postoperative patients can achieve the effect of complete urinary control. However, after surgery, most patients still have a small amount of urine leakage and need to use pads. It is worth noting that most studies did not consider the severity of SUI. Previous studies have found that [2] the severity of urinary incontinence will affect the effect after implantation, and most of the literature did not analyze the clinical cases of different degrees of urinary incontinence. In this study, subgroup analysis was performed on the dry rate and social dry rate according to the severity of urinary incontinence. The results showed that the heterogeneity improvement was not obvious, suggesting that subgroup analysis according to the severity of urinary incontinence could reduce heterogeneity. However, it was not necessarily the main source of heterogeneity.

**Daily urine pad count.**   The purpose of AUS implantation is to obtain urinary control ability to achieve complete urinary control or social urinary control standards. In this meta-analysis, the follow-up data of daily pad counts were compared with baseline data. The study showed that the number of pads used per day after AUS was significantly reduced by about 4

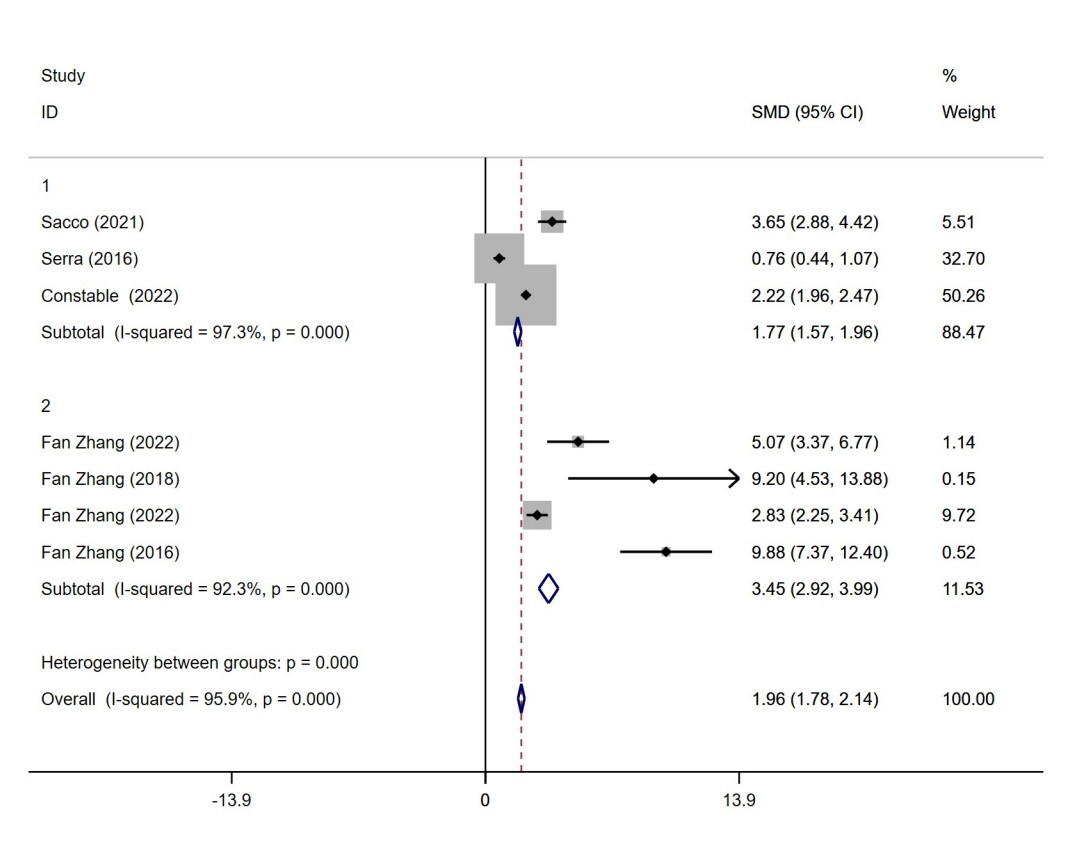

**Fig 5. Meta-analysis for quality of life score for patients treated with AUS.**

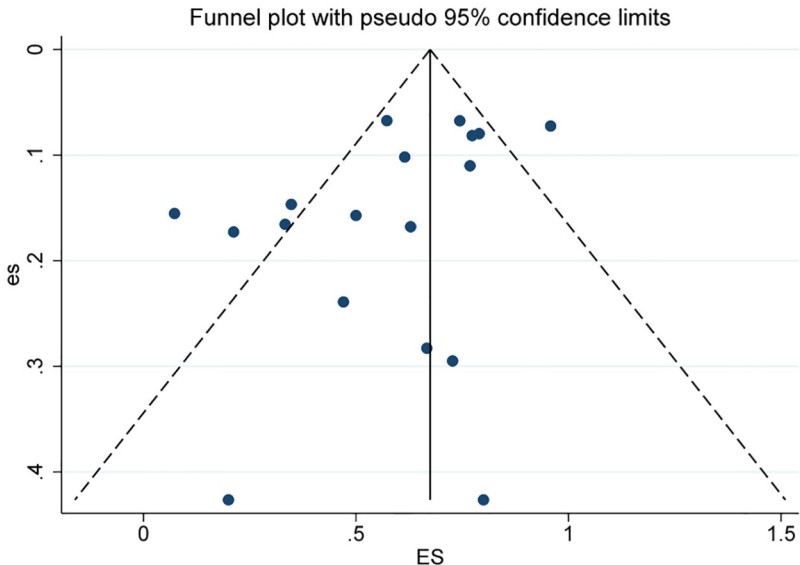

**Fig 6. Funnel plot for publication bias for dry rate n (%) for patients treated with AUS.**

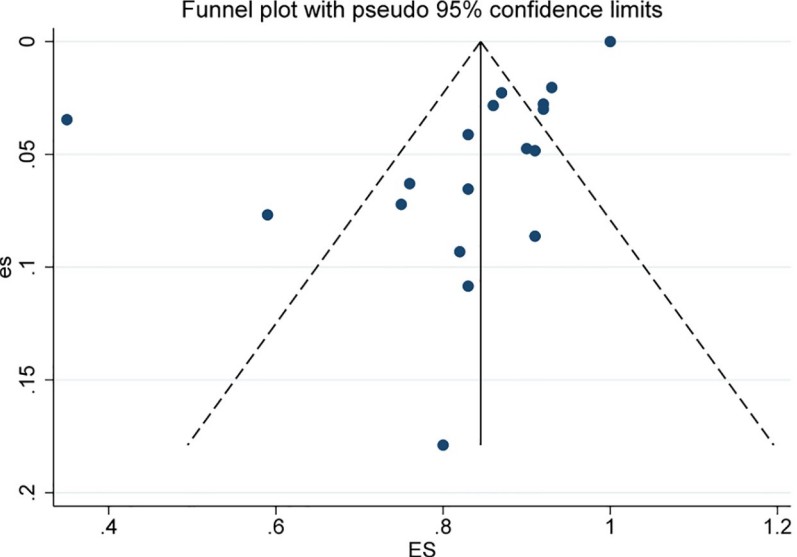

**Fig 7. Funnel plot for publication bias for social dry rate n (%) for patients treated with AUS.**

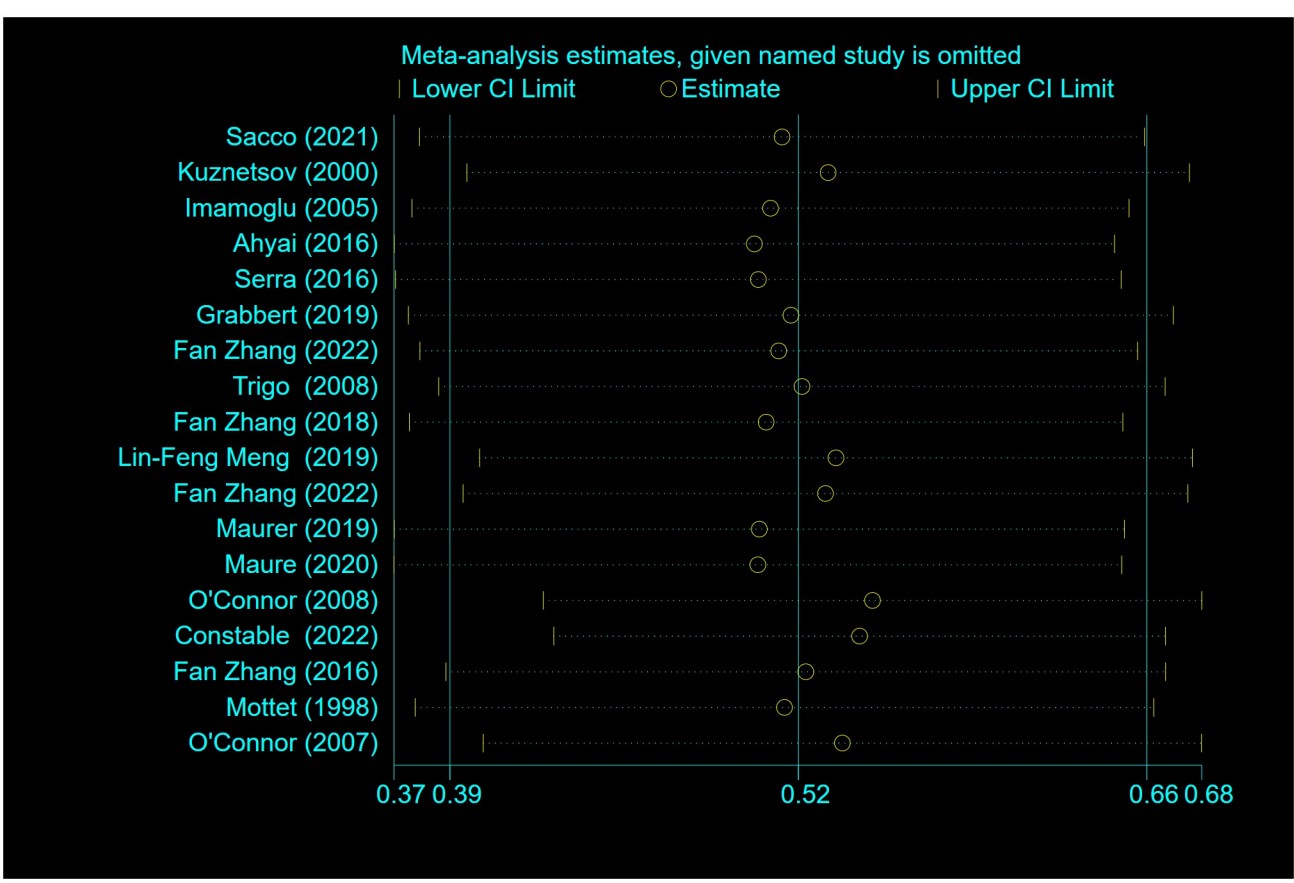

**Fig 8. Sensitivity-analysis for dry rate n (%) for patients treated with AUS.**

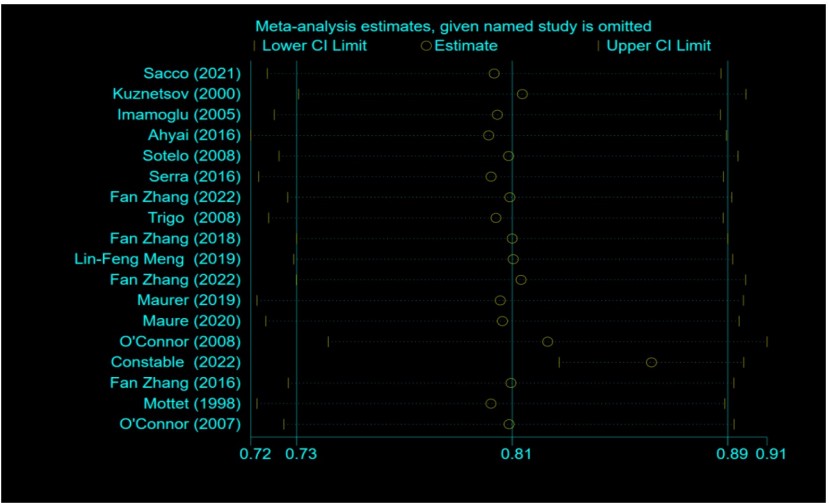

**Fig 9. Sensitivity-analysis for social dry rate n (%) for patients treated with AUS.**

compared with that before the intervention, indicating that AUS could significantly reduce the number of pads used by patients. However, due to the different types of pads currently available on the market, the pad count included in the study may have some errors in the assessment of urinary incontinence [32]. In addition to the use of urine pad evaluation, the urine pad test is more accurate and objective, but due to the lack of data, data analysis cannot be performed. Therefore, more systematic and accurate methods are needed to improve the understanding of the efficacy of AUS.

**Quality of life.** SUI after prostate surgery is a severe complication, and the impact of AUS on quality of life is one of the key indicators to measure treatment success. Minato et al.conducted a longitudinal study on patients after AUS implantation and found that urinary incontinence and ICIQ-SF scores significantly improved immediately after surgery. The preoperative ICIQ-SF score was 18.9 and decreased to 4.5 one month after AUS activation [33]. The research results are consistent with this paper. Among the 7 studies that met the inclusion criteria in this article published preoperative and postoperative quality of life data

**Table 4. GRADE evidence summary table.**

| Outcomes | Inconsistency | Indirectness | Imprecision | Risk of bias | Other consideration | Effect(95%CI) | No. of Participants (Studies) | Quality of the Evidence (GRADE) |
|---|---|---|---|---|---|---|---|---|
| Dry rate | Serious[1] | Not serious | Serious[3] | Not serious | None | RR = 0.52,95%CI (0.39~0.66) | 1188(18) | Very Low |
| Social dry rate | Serious[2] | Not serious | Not serious | Not serious | None | RR = 0.81,95%CI (0.73~0.89) | 1201(18) | Moderate |
| Pads use (pads/day) | Serious[2] | Not serious | Not serious | Not serious | None | SMD = 2.68,95%CI (2.07~3.29) | 646(9) | Moderate |
| Quality of life | Serious[2] | Not serious | Serious[4] | Not serious | None | SMD = 1.96,95%CI (1.78~2.14) | 387(7) | Low |

[1]. The overlap is not good, $I^2 > 50\%$.

[2]. $I^2 > 50\%$.

[3]. very wide confidence intervals.

[4]. few participants.

[12, 17 22, 23, 25, 27, 29].3 of them used ICIQ-SF score to prospectively analyze the quality of life outcomes of 307 SUI patients after surgery [12, 17, 29]. The patient's quality of life score was about 17.4 points before surgery and decreased to about 5 points after surgery.4 studies retrospectively analyzed the quality of life outcomes of 80 postoperative SUI patients [22, 23, 25, 27] using the VAS score, and the score decreased from about 7.6 to about 2.4. The research indicates that although a significant proportion of men still experience some degree of UI after AUS implantation, the majority of male patients show significant improvement in urinary incontinence, suggesting a notable enhancement in the quality of life for male patients following AUS surgery.

## Limitations of research analysis

There are some limitations in the analysis of this paper, and these problems should be considered when analyzing the results. ①A total of 19 articles, 12 retrospective cohort studies, 6 prospective cohort studies, and 1 randomized controlled trial was included in this study, with different research types and designs. Research design issues reflect the limited quality of studies available in our meta-analysis.②The follow-up time of each study was inconsistent. The follow-up time of some studies was only 12 months, and the longest follow-up time was only 85 months.③There are also differences in the operation methods of different surgeons, such as the timing of implantation, incision design, and the choice of suture methods. These differences may affect the results.③④At present. There are no standardized evaluation criteria to evaluate the outcome of SUI after prostate surgery, including the definition of urinary incontinence, the standardized evaluation of dry rate and social dry rate, which may confuse the final results of the study.③⑤Due to the lack of data, the study did not assess the incidence of complications.

## Conclusion

Based on the results of this study, the use of AUS can effectively treat SUI after prostate surgery, thereby significantly reducing the number of urinary pads used by patients every day and improving the quality of life of patients. However, due to the limitations mentioned above in this study, more high-quality, long-term, high-quality studies are still needed to draw more accurate and reliable conclusions for the clinical application of AUS under the analysis of multi-center and large samples.

## Supporting information

**S1 Checklist. PRISMA 2020 checklist.**
(DOCX)

**S1 Table. The summarized results of quality assessment for insuifluded studies.**
(DOCX)

**S2 Table. Search strategy.**
(DOCX)

**S3 Table. Raw data included in the study.**
(XLSX)

**S4 Table. Raw data of dry rate.**
(XLSX)

**S5 Table. Raw data of social dry rate.**
(XLSX)

**S6 Table. Raw data for daily pad use.**
(XLSX)

**S7 Table. Raw data of quality of life.**
(XLSX)

## Author Contributions

**Data curation:** Qin Yang.

**Investigation:** Yue Li.

**Software:** Yue Li.

**Writing – original draft:** Yue Li.

**Writing – review & editing:** Yue Li, Xiao Li.

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
