## [Decision Letter · Decision Letter 0]

18 Jul 2023

PONE-D-23-15504Effectiveness of Artificial Urinary Sphincter to Treat Stress Incontinence after Prostatectomy: A Meta-Analysis and Systematic ReviewPLOS ONE

Dear Dr. Li,

Thank you for submitting your manuscript to PLOS ONE. After careful consideration, we feel that it has merit but does not fully meet PLOS ONE’s publication criteria as it currently stands. Therefore, we invite you to submit a revised version of the manuscript that addresses the points raised during the review process.

We look forward to receiving your revised manuscript.

Kind regards,

Marcelo Chen

Academic Editor

PLOS ONE

3. Please remove your figures from within your manuscript file, leaving only the individual TIFF/EPS image files, uploaded separately. These will be automatically included in the reviewers’ PDF.

Reviewers' comments:

Reviewer's Responses to Questions

**Comments to the Author**

1. Is the manuscript technically sound, and do the data support the conclusions?

Reviewer #1: Yes

Reviewer #2: Yes

2. Has the statistical analysis been performed appropriately and rigorously? 

Reviewer #1: Yes

Reviewer #2: Yes

3. Have the authors made all data underlying the findings in their manuscript fully available?

Reviewer #1: No

Reviewer #2: Yes

4. Is the manuscript presented in an intelligible fashion and written in standard English?

Reviewer #1: Yes

Reviewer #2: No

5. Review Comments to the Author

Reviewer #1: This is a well-written systematic review and meta-analysis on data about objective and patient-reported efficacy of the artiphicial urinary sphincter for the treatment of male stress urinary incontinence. Unfortunately, and expectedely, the main limitation of the meta-analysis is the high heterogeneity that decreases the level of the evidence of the results. Furthermore, a GRADE evaluation (the most used approach to rating the certainty of evidence in systematic reviews and meta-analysis) is missing. However, the summary results remain of great interest.

Minor comments:

- it semms to me that to few studies have been retrieved from the searhed database, for instance, only 20 studies were found in PubMed, although the search terms were quite broad, please explain better the search strategy.

- are the author sure to have not included duplicated studies/patients populations? there are four studies by Fan Zhang.

- table 3 can be included as supplementary material, instead, a table with more detailed study characteristics is missing (e.g., patients enrolled and patients analyzed with lost to follow-up, type of sphincter, intervention technique, case mix such as irradiated patients or redo procedure/salvage procedure, ...)

- for publication bias analysis at least ten articles need to be included in the meta-analysis and this hold true for dryness and social dryness outcomes, however, only social dryness was tested, please explain.

- sensitivity analysis as well was performed for social dryness only.

- the inclusion of a GRADE evaluation would increase greatly the quality of the paper.

Reviewer #2: Dear authors,

I congratulate to the systematic review and meta-analysis.

There are some one the same subject, but already older.

I have some comments to made:

Methods

Instead of „Survey“, would cohort study be more accurate for description? It implies observational design.

Case: do you mean case report?

We’re there any restrictions in follow-up time?

Results:

Drying rate: do you mean dry rate?

Table 3 is very difficult to read. Suggest to take into consideration different presentation, rather this table. Certainty of evidence is usually presented in an image in addition.

Actually there is many literature with large cohort from majo clinics, for example, and the domino group (Kretschmer/Hüsch), which I am missing. Please explain.

Suggest to add the median follow-up time of the analyses.

6. PLOS authors have the option to publish the peer review history of their article (what does this mean?). If published, this will include your full peer review and any attached files.

Reviewer #1: **Yes: **EMILIO SACCO

Reviewer #2: No

---

## [Author Response · Author response to Decision Letter 0]

10 Aug 2023

Response to reviewers

Revision report

First of all, I would like to express our sincere gratitude to the reviewers for their comments. These comments are all valuable and helpful for revising and improving our manuscript, as well as the important guiding significance to our research. We have studied comments carefully and have made correction which we hope to meet with approval. Revised portions are marked in red in the revised version. All page numbers refer to the revised manuscript file with tracked changes.The responses to the reviewer's comments are listed below.

Responses to reviewers (original comments by reviewers are in blue color)

Reviewer # 1:

3.Comment: Have the authors made all data underlying the findings in their manuscript fully available?

3.Reply:Thank you for your comment.We feel sorry for the problems.We have uploaded the data related to the research results as supporting information(S1-S7). All data are public and shared.

5.a)Comment: it semms to me that to few studies have been retrieved from the searhed database, for instance, only 30 studies were found in PubMed, although the search terms were quite broad, please explain better the search strategy.

5.a)Reply:We are very sorry for our negligence.As suggested by the reviewer,we identified all potential studies using combinations of the following terms (free text and/or medical subject headings (MeSHI) terms adapted to the requirements of each database).Taking pubmed search as an example: (("urinary sphincter, artificial"[MeSH Terms] OR "artificial urinary sphincter"[Title/Abstract] OR (("Sphincter"[All Fields] OR "sphincter s"[All Fields] OR "sphincteral"[All Fields] OR "sphincteric"[All Fields] OR "Sphincters"[All Fields]) AND "artificial urinary"[Title/Abstract]) OR (("urinary tract"[MeSH Terms] OR ("Urinary"[All Fields] AND "tract"[All Fields]) OR "urinary tract"[All Fields] OR "Urinary"[All Fields]) AND "sphincters artificial"[Title/Abstract])) AND "urinary incontinence"[MeSH Terms] AND "male"[MeSH Terms]) AND (clinicaltrial[Filter] OR randomizedcontrolledtrial[Filter])The full search strategy is detailed in S2 Tables. 

5.b)Comment: are the author sure to have not included duplicated studies/patients populations? there are four studies by Fan Zhang.

5.b)Reply:We thank the reviewer for raising this question.For the 4 studies of Fan Zhang, we re-read the literature again and determined that the repeated studies / patient population were not included.

5.c)Comment: table 3 can be included as supplementary material, instead, a table with more detailed study characteristics is missing (e.g., patients enrolled and patients analyzed with lost to follow-up, type of sphincter, intervention technique, case mix such as irradiated patients or redo procedure/salvage procedure, ...)

5.c)Reply:We agree with you that table 3(The summarized results of quality assessment for insuifluded studies) can be included as supplementary material, We have uploaded the table as a supplementary material. The details of Quality assessment of included studies are shown in Additional S1 Tables.

Secondly, for the problem that the content of the study characteristics table is not detailed enough, we follow your recommendations and Add the lost to follow-up rate, intervention techniques, radiotherapy patients, re-do procedures, etc. Because the type of sphincter studied in this paper has been mentioned in Table 1, the type of sphincter is artificial urinary sphincter AMS 800 TM ( Boston Scientific, Boston, USA ), so the type of sphincter is not included in the study characteristics table. In addition to adding the above content. We also add Study design, Complication rate, UI surgery history.Study characteristics table（Table 3 Summary of basic characteristics and outcomes of selected studies for meta-analysis) supplementary items are marked in red on pages 7-9 of the manuscript. And on page 6, We briefly summarize the supplementary items in paragraph 2 of section 2 ( results ):“The main intervention technique was single / double-cuff AUS.…,The lost to follow-up rate was between 2.6 % and 16 % …,Patients with re-do procedures ranged from 2% to 21%.( Table 3 )”

5.d)Comment: for publication bias analysis at least ten articles need to be included in the meta-analysis and this hold true for dryness and social dryness outcomes, however, only social dryness was tested, please explain.

5.d)Reply:We are very sorry for our careless mistakes.There is indeed a lack of publication bias analysis of dry rate in the manuscript.We have added the suggested content to the manuscript on page 14~15，We added the Egger test of the dry rate and analyzed the cause of the publication bias on page 14：“Therefore…the different proportion of patients after previously failed anti-incontinence devices and the different proportion of patients receiving radiation likely explain the publication bias situation observed”. On page 15, we add a funnel plot of the dry rate, as shown in Fig 6 Funnel plot for publication bias for dry rate n (%) for patients treated with AUS.

5.e)Comment: sensitivity analysis as well was performed for social dryness only.

5.e)Reply:Thank you very much for the reviewer 's opinion. As the reviewer said, the sensitivity analysis of the dry rate is missing in the manuscript. According to the reviewer 's opinion, we added the sensitivity analysis of the dryness rate on page 16 of the manuscript, as shown in Figure 8.

5.f)Comment: the inclusion of a GRADE evaluation would increase greatly the quality of the paper.

5.f)Reply:We think this is an excellent suggestion, and the inclusion of GRADE evaluation can greatly improve the qsuality of the article.According to the reviewer 's suggestion, we added GRADE evaluation to the manuscript. First, in the method section of the abstract on page 1, we propose to use the GRADE guidelines to evaluate the overall evidence:“And methodologic quality of the overall body of evidence was evaluated using the GRADE (Grading of Recommendations Assessment, Development, and Evaluation) guidelines.” Secondly, in the results section of the summary on page 2, we summarized the levels of evidence for dry rate, social dry rate, Pads use ( pads / day ), and quality of life.“The evidence level of GRADE of dry rate is very low, …，and the evidence level of Quality of life is low.” Then, in the method section on pages 4-5, the content and level of GRADE assessment are supplemented and described:“GRADE ( Grading of Recommendations Assessment, Development and Assessment ) .… ,Confidence of the effect estimates were described as high, moderate, low and very low .(Table 2)”We also supplemented :Table 2. GRADE Working Group grades of evidence. Finally, in the last part of the results on pages 16-17, we supplemented :Rating the quality of evidence, We briefly describe the level of quality of evidence“The evidence level of GRADE of Dry rate is very low, …,The reasons for the downgrade are as follows.( table 4 )”. And list Table 4. GRADE evidence summary table.

Reviewer # 2:

4.Comment:Is the manuscript presented in an intelligible fashion and written in standard English?

4.Reply:We apologize for the poor language of our manuscript. We worked on the manuscript for a long time and the repeated addition and removal of sentences and sections obviously led to poor readability. We have now worked on both language and readability and have also involved native English speakers for language corrections. We really hope that the flow and language level have been substantially improved.

5.a)Comment:Methods

Instead of „Survey“, would cohort study be more accurate for description? It implies observational design

5.a)Reply:We sincerely thank the reviewer for careful reading. As suggested by the reviewer, we have corrected the “Survey” into “cohort study”.

5.b)Comment:Methods

Case: do you mean case report?

5.b)Reply:We are very sorry because the language of the manuscript is ambiguous and has caused you a misunderstanding. On page 3, according to your recommendations, we have revised the "prospective case series" into a "prospective cohort study".

5.c)Comment:Methods

We’re there any restrictions in follow-up time?

5.c)Reply:Thanks for the reviewer 's opinion. After literature screening, the number of studies that met the inclusion criteria was small, so there was no restriction on the follow-up time. In this case, we hope the reviewer understands. In the future, we will consider the inclusion of follow-up time in the manuscript to reduce the impact of follow-up time differences on the results.

5.d)Comment:Results

Drying rate: do you mean dry rate?

5.d)Reply:Thank you for pointing out the issue. We have noticed the inconsistency in the translation of "dry rate" throughout the manuscript. We sincerely apologize for this mistake and will make every effort to rectify it. Based on your feedback, we have amended "Drying rate" to "dry rate" in the manuscript.

5.e)Comment:Results

Table 3 is very difficult to read. Suggest to take into consideration different presentation, rather this table. Certainty of evidence is usually presented in an image in addition.

5.e)Reply:Thank you for your feedback. We acknowledge that the current format of Table 3 may be difficult to read. For your suggestion, since the modified 18-item Delphi checklist does not have software that can form images, we use Table 3(The summarized results of quality assessment for insuifluded studies) as a supplementary material, the details of Quality assessment of included studies are shown in Additional S1Table.

5.f)Comment:Actually there is many literature with large cohort from majo clinics, for example, and the domino group (Kretschmer/Hüsch), which I am missing. Please explain.

5.f)Reply:Thank you for your comment and bringing up the missing literature sources from major clinics, such as the studies conducted by the Kretschmer/Hüsch group. In response to your concern, we searched 268 articles of Kretschmer and 64 articles of Hüsch, we have thoroughly reviewed the literature again. Because the above articles did not meet the inclusion criteria, the study of the two authors was not included. In addition to the studies of the above two authors, we have thoroughly reviewed the literature again. The reviewed studies were excluded because they did not meet the inclusion criteria of manuscripts.

5.g)Comment:Suggest to add the median follow-up time of the analyses.

5.g)Reply:Thank you for your suggestion, we have added the median follow-up time to (Line 3, page 6).

Thank the reviewers for their valuable comments and suggestions on our research. We have carefully considered each of your suggestions and made corresponding changes in the revised version.I look forward to-hearing from you.

Yours sincerely,

Xiao Li

---

## [Editor Report · Decision Letter 1]

21 Aug 2023

Effectiveness of Artificial Urinary Sphincter to Treat Stress Incontinence after Prostatectomy: A Meta-Analysis and Systematic Review

PONE-D-23-15504R1

Dear Dr. Li,

We’re pleased to inform you that your manuscript has been judged scientifically suitable for publication and will be formally accepted for publication once it meets all outstanding technical requirements.

Kind regards,

Marcelo Chen

Academic Editor

PLOS ONE

---

## [Editor Report · Acceptance letter]

24 Aug 2023

PONE-D-23-15504R1 

Effectiveness of Artificial Urinary Sphincter to Treat Stress Incontinence after Prostatectomy: A Meta-Analysis and Systematic Review 

Dear Dr. Li:

I'm pleased to inform you that your manuscript has been deemed suitable for publication in PLOS ONE. Congratulations! Your manuscript is now with our production department. 

Kind regards, 

on behalf of

Dr. Marcelo Chen 

Academic Editor

PLOS ONE